# In Vitro Production of Embryos from Prepubertal Holstein Cattle and Mediterranean Water Buffalo: Problems, Progress and Potential

**DOI:** 10.3390/ani11082275

**Published:** 2021-08-01

**Authors:** Luke Currin, Hernan Baldassarre, Vilceu Bordignon

**Affiliations:** Department of Animal Science, McGill University, Sainte-Anne-de-Bellevue, QC H9X 3V9, Canada; luke.currin@mail.mcgill.ca (L.C.); hernan.baldassarre@mcgill.ca (H.B.)

**Keywords:** Holstein, Mediterranean Water Buffalo, in vitro embryo production, laparoscopic ovum pickup, accelerated genetic gain, prepubertal, embryo development

## Abstract

**Simple Summary:**

In vitro embryo production using oocytes from prepubertal cattle and buffalo collected by laparoscopy can be used to produce embryos from genetically superior females. Following transfer of these embryos into adult recipient animals, multiple offspring can be produced from these elite animals in a very short timeframe, long before they reach sexual maturity, thereby reducing the generation interval and accelerating genetic gain. This review article summarizes recent advances in this technology, outlines the current limitations, and suggests possible avenues to further improve this emerging biotechnology.

**Abstract:**

Laparoscopic ovum pick-up (LOPU) coupled with in vitro embryo production (IVEP) in prepubertal cattle and buffalo accelerates genetic gain. This article reviews LOPU-IVEP technology in prepubertal Holstein Cattle and Mediterranean Water Buffalo. The recent expansion of genomic-assisted selection has renewed interest and demand for prepubertal LOPU-IVEP schemes; however, low blastocyst development rates has constrained its widespread implementation. Here, we present an overview of the current state of the technology, limitations that persist and suggest possible solutions to improve its efficiency, with a focus on gonadotropin stimulations strategies to prime oocytes prior to follicular aspiration, and IVEP procedures promoting growth factor metabolism and limiting oxidative and endoplasmic reticulum stress.

## 1. Introduction

In vitro embryo production (IVEP) and embryo transfer (ET) technologies have had a momentous impact on livestock production, with their use growing substantially in recent years. Despite barely being used on a commercial scale as recently as the late 1990s, IVEP has increased at an average annual rate of 12%, according to data provided by the International Embryo Transfer Society [1,2]. Moreover, it has been applied in most important livestock species, as reviewed in previous publications, e.g., cattle [2], buffalo [3], camelids [4], swine [5], goat and sheep [6,7], and cervids [8]. In cattle, where IVEP is broadly used, the majority of embryos transferred worldwide have been produced in vitro since 2016 [1]. Although no single factor can be attributed as the sole cause of this major milestone, improved media composition, the introduction of sexed semen, faster turnover compared to conventional multiple ovulation embryo transfer (MOET), and the ability to use semen from multiple bulls on oocytes from a single donor at the same time are all believed to be contributing factors [9].

Another key factor that explains IVEP expansion is the refinement of technologies to enable safe and practical collection of oocytes from live females. In large adult animals, most oocytes used for commercial embryo production are collected via ultrasound-guided trans-vaginal ovum pickup (OPU). However, in species that are too small for oocyte collection via OPU (e.g., sheep, goat, deer), a laparoscopic ovum pick-up (LOPU) procedure was developed in the early 90s [10]. Since then, it has been refined and adapted for use in a wide range of both domestic and wild species [10,11,12,13,14,15,16,17,18]. The LOPU approach has several advantages over OPU, including that the ovary is viewed directly with a depth of field, rather than on a two-dimensional sonogram, enabling superficial follicles to be aspirated accurately without risking injury to the ovarian stroma [19]. This minimizes ovarian trauma, and hence the risk of sequels including tissue adhesions. As such, LOPU can be repeated on a regular basis while minimizing long-term reproductive concerns [11,20].

Of particular interest is the application of LOPU to conduct IVEP in very young animals. LOPU allows the recovery of oocytes from animals as young as two months of age, long before they are sexually mature or large enough for ultrasound guided OPU. Subsequently, IVEP allows for these oocytes to be fertilized in vitro to produce blastocysts, which are then transferred into adult recipient females, as shown in Figure 1. Using this approach, multiple offspring from the donor animal can be born before it reaches sexual maturity. Using LOPU-IVEP, it is now possible to exploit the large ovarian pool of oocytes present at young ages to rapidly proliferate genetically superior, valuable, or endangered animals [19,20]. It also provides a faster mechanism for the proliferation of animal lineages of particularly valuable genotypes [21]. Additionally, from a more basic-science perspective, prepubertal animals are also excellent negative models for the acquisition of developmental competence, leading to a better understanding of infertility and the development of new fertility treatments [22].

There are two main reasons for the interest in using prepubertal animals as oocyte donors. First, the ovarian pool of available oocytes is vast; prepubertal animals consistently yield large numbers of cumulus-oocyte complexes (COCs) compared to their adult counterparts [19,23]. Second, early propagation of elite animals results in shorter intervals between generations, thereby increasing the rate of genetic gain [24] and enabling faster access to the latest genetic lineages. However, multiple studies have consistently shown that, although large number of COCs can be recovered, poor embryo development rates result in few blastocysts from prepubertal-derived oocytes in many domestic livestock species including cattle [25,26], buffalo [27] goat [28,29], sheep [30,31], and pig [32,33]. Although differences in oocyte competence vary among species, in cattle, prepubertal oocytes typically yield a 10–15% blastocyst rate compared to ~30% using oocytes from adult animals [23]. While the exact reasons for the impaired competence are unknown and are most likely a combination of multiple factors, various differences have been noted such as smaller oocyte size, incomplete cytoplasmic maturation, variations in gene expression, and alterations in protein synthesis and metabolism [26,34,35].

This review will focus on prepubertal reproductive technologies, sometimes dubbed ‘juvenile in vitro embryo transfer’ (JIVET), in Holstein–Friesian cattle (*Bos taurus taurus*) and Mediterranean water buffalo (*Bubalus bubalis*). Together, these species serve as complementary animal models to investigate prepubertal oocyte competence and improve prepubertal reproductive technologies since Holsteins mature relatively quickly while water buffalo mature much more slowly. In normal breeding practices using artificial insemination, Holstein heifers typically give birth to their first calf around two years of age, while, on average, water buffalo heifers are not expected to calf until around three years of age. As such, the goals of this review are to outline the current state of the technology, identify research gaps and suggest possible future avenues of research.

## 2. Increasing the Rate of Genetic Gain by Shortening Generation Intervals

Selective breeding, or artificial selection, is the practice where individuals are bred based on specific merits in order to proliferate a desirable trait. Broadly speaking, exceptional animals are bred to produce superior offspring. Recently, genomics has revolutionized selective breeding strategies and reliable single-nucleotide polymorphisms for various traits have been identified in both cattle and water buffalo [36]. The rate at which these genetic gains (the difference in genetic value between parent and offspring) take place is inversely correlated with the generation interval [37]. Therefore, it is beneficial to breed the best animals at the youngest age possible in order to maximize the rate of genetic gain. Using buffalo as an example, if a calf undergoes LOPU/IVEP/ET at two months of age, offspring would be born at around the time the donor animal is one year old, effectively decreasing the generation interval by up to two years.

## 3. History of LOPU-IVEP in Prepubertal Calves

It was identified early on that using prepubertal animals in breeding programs would lead to dramatic increases in the rate of genetic gain. However, most early attempts at using MOET in prepubertal cattle predominantly failed. Some of the first attempts in the early 1970s noted that, when embryos sourced from prepubertal animals were placed in culture, development arrested before reaching the morula stage [38,39,40]. It should be noted that IVEP technology was still in its infancy at that time. Nevertheless, these pioneering studies showed that prepubertal animals could respond to exogenous gonadotropin stimulation [41]. Although animals responded well to follicle-stimulating hormone (FSH) treatment, they did not ovulate reliably in response to injections of pituitary extracts high in luteinizing hormone (LH) [41,42], resulting in low recovery rates and poor embryonic development [38,39]. Based on those observations, assumptions were made that the prepubertal reproductive tract was detrimental and not conducive to normal fertilization and early embryo development [38,41]. Ultimately, this resulted in MOET strategies being abandoned and attention instead turned to LOPU-IVEP.

Substantial research was done in the 1990s to develop reliable LOPU and IVEP techniques for prepubertal animals in several species. Studies during this period showed that the LOPU component was largely successful, but IVEP was not. Following LOPU, multiple authors reported high oocyte yields in young animals, often more than what is typically recovered from adult animals [19,23,43]. Following IVEP, the oocytes from young animals resulted in blastocyst development rates that were consistently lower than rates in mature animals [25,44,45,46,47]. For example, Revel and associates found similar fertilization and cleavage rates between oocytes from three-month-old heifer calves and adult cattle, but prepubertal oocytes failed to produce similar blastocyst rates [23]. These poor results, combined with the inability to identify genetically superior animals at such young ages at that time, led to the loss of interest in prepubertal LOPU-IVEP research projects for around 20 years.

Since these studies in the 1990s, significant advancements in marker-assisted selection, genomics, and IVEP have renewed interest and demonstrated the potential to circumvent many of the initial shortfalls. Genomic marker selection in particular is having a huge impact on the dairy industry where the production phenotype can be accurately predicted as soon as the animal is born through screening of single-nucleotide polymorphisms [48,49]. Along with the progress in genome selection, significant improvements in IVEP practices have been accomplished in recent years, resulting in the ability to produce high quality embryos in vitro, comparable to their in vivo derived counterparts [9]. Recent innovations such as sequential media compositions and advanced low-oxygen tension incubators have enabled production of embryos in vitro possessing cryotolerance capabilities similar to embryos produced in vivo [2,50]. As genomic selection and modern IVEP technology become more and more cost-effective in the future, their application and use are expected to continue growing.

Despite significant progress in recent years, problems with prepubertal IVEP technologies still exist. As evidenced from studies in different species, prepubertal oocytes have a reduced developmental competence compared to adult oocytes, with fewer IVEP embryos reaching the blastocyst stage, as observed in bovine [23], buffalo [27], ovine [30], caprine [51], and swine [52]. It has been shown that calves respond well to FSH stimulation and produce many follicles, often producing more than cows [19,23,43]. Consequently, prepubertal donors typically produce more COCs and 2-cell stage embryos than adult donors. Blastocysts derived from prepubertal oocytes are competent to support full-term development and normal offspring have been produced in multiple species, including buffalo [7,27] and cattle [23,53,54]. Hence, the primary challenge remains the improvement of oocyte competence to enable higher embryo development rates to the blastocyst stage. In this regard, learning how to prime and prepare prepubertal oocytes, both in vivo inside the follicle and in vitro during maturation and culture, seems the most logical and promising path to consolidate LOPU-IVEP uses in prepubertal breeding schemes, as shown in Figure 2.

## 4. Understanding Developmental Competence of Oocytes

One of the greatest challenges in overcoming the impaired developmental competence of prepubertal oocytes is that the underlying reasons are not fully understood, and the cause is most likely a combination of multiple factors. For example, the hypothalamic-pituitary-ovarian axis in prepubertal animals is immature, which could lead to defective signalling and steroidogenesis in ovarian follicles. In turn, an improper follicular micro-environment could affect metabolism within the oocyte itself or the crosstalk between the oocyte and granulosa cells, ultimately resulting in oocytes unable to reach full developmental competence.

### 4.1. The Hypothalamic–Pituitary–Ovarian (HPO) Axis

The HPO axis is essential for the management of the oestrous cycle and, consequently, fertility. Kisspeptins (*Kp*) are a family of neuropeptides in the hypothalamus, which were discovered in 2003 to operate upstream of gonadotropin-releasing hormone (GnRH) signalling [55]. GnRH neurons express the receptor for kisspeptin, GPR54, and consequently have been implicated in many critical roles including timing the onset of puberty, secretion of gonadotropins, transmission of the negative and positive feedback loops, and generation of the LH surge [56,57].

This upstream hypothalamus signalling is believed to be the last component of the HPO axis to mature in juvenile heifers, and is the limiting factor determining the HPO functionality prior to puberty [58]. Specifically, the number of *Kp*-positive cells in the arcuate nucleus and pre-optic area are believed to be responsible for the negative and positive feedback loops, respectively, and have been shown to increase during prepubertal development in the ewe [59]. Downstream, in the pituitary, GnRH receptors do not change with age, and secrete gonadotropins in response to GnRH at a very young age [60,61]. In the ovary, the relative mRNA abundance of FSH receptor in granulosa cells is significantly lower in prepubertal Holsteins compared to adult cows, possibly explaining the smaller average follicle size in prepubertal animals, and consequently the reduced developmental competence of oocytes [62].

### 4.2. Follicular Microenvironment

The lower developmental potential of calf oocytes may be due to environmental deficiencies in vivo prior to retrieval [23,63]. Hence, a clear understanding of the follicle and its follicular fluid is important. Calf follicular fluid contains approximately half the LH concentration compared to cow follicular fluid (2.0 ± 0.2 ng/mL vs. 4.0 ± 0.3 ng/mL) [64]. This is in accordance with the plasma concentration of LH, which is also lower in younger animals [65]. Although changes in LH concentration may have no direct impact on the oocyte itself due to a lack of LH receptors, it would affect steroidogenesis and androgen production in granulosa and theca cells [66]. A disruption in estrogen production would affect the transcription of genes regulated by estrogen response elements. Alternately, impaired androgen metabolism could also affect fertility, as androgen-receptor knock-out mice are sub-fertile [67]. In a similar manner to LH, calf follicular fluid has also been shown to contain approximately half the estradiol content compared to adults (6.3 ± 2.1 ng/mL vs. 12.7 ± 5.5 ng/mL) [64]. Collectively, it can be speculated that these differences in the follicular micro-environment may negatively impact the acquisition of developmental competence, and may partially explain the low IVEP outcomes observed in calves [63]. This further emphasizes the importance of suitable gonadotropin stimulation regimes to emulate a follicular microenvironment that will promote oocyte competence prior to LOPU.

### 4.3. Oocyte and Granulosa Cell Crosstalk

Oocyte competence is dependent on intercellular communication within the ovarian follicle during follicular growth and development, and is regulated by endocrine, paracrine, and autocrine factors [68]. While direct inter-cellular connections are mediated via gap junctions and transzonal projections (TZPs) [69], indirect intercellular communication can occur through extracellular vesicles (EVs) secreted into the follicular fluid [70]. Collectively, these pathways facilitate bi-directional communication, signaling and transport of molecules between the oocyte, granulosa, and theca cells [70,71].

Developmental competence increases gradually and sequentially as oocytes increase in size due to transcriptional activity during follicular and oocyte growth [72,73]. This is vital as oocytes from prepubertal animals are smaller and have a thinner zona pellucida than those from adults, despite originating from follicles of the same size [74]. For example, calf oocytes have a mean diameter of 118.04 ± 1.15 μm compared to a mean diameter of 122.83 ± 0.74 μm for mature cows [24,74]. Since a small variation in diameter represents a larger variation in volume, small variations in diameter may have important impacts on developmental competence. As such, the capacity of bovine oocytes to mature to metaphase II during IVM is positively correlated with their diameter [75]. Aside from diameter, several cytoplasmic differences have also been observed between oocytes from prepubertal and adult animals. For example, oocytes from adult cows have more lipid droplets in their cytoplasm compared to those from heifers, both before and after IVM [63]. Other differences include incomplete cytoplasmic maturation, altered gene expression and protein synthesis, as well as defective metabolism in oocytes from young animals [26,34,35].

More recently, the intimate relationship between the oocyte and cumulus cells has been investigated to better define the role of TZPs [76,77]. Although more research needs to be done to determine how the physiology, distribution and retraction of TZPs impacts IVEP outcomes in both prepubertal and adult oocytes, TZPs are known to facilitate communication and the transport of essential molecules between granulosa cells and the oocyte [76,78]. Despite differences observed in the organization of TZPs in COCs from lambs compared to adult ewes, the impact on embryo development remains unclear [79].

In addition to intercellular communication via TZPs, the roles of EVs on intra-follicular cell communication has also become of particular interest [70]. EVs are small lipid bilayer particles secreted by cells into the extracellular space, which then diffuse and act on secondary target cells, transporting various molecules including proteins, lipids, messenger RNA (mRNA), and microRNA (miRNA) [80,81]. Since the initial discovery of EVs in equine follicular fluid in 2012 [82], they have since been described in bovine [83] and porcine follicular fluid [84,85] and were shown to play multiple roles inside the follicle, including granulosa cell proliferation and cumulus expansion [86,87]. Notably, studies have found variability in EV and miRNA profiles when comparing follicular fluid from follicles of different sizes and young vs. old animals [82,86,87,88]. For example, da Silveira found significant differences in the number and profiles of miRNAs present when comparing follicular fluid from young (3–13 y.o.) and old (>20 y.o.) mares [82,89]. Others have found similar results when comparing younger (<31 y.o.) and older (> 38 y.o.) women [90]. How these findings may translate into prepubertal vs. adult cattle and buffalo remains unknown. However, it has been shown that supplementation with EVs in vitro was able to increase blastocyst rates in cattle to 37%, compared to 26% using IVM with EV-free fetal calf serum [91]. Thus, it is possible that supplementation with adult EVs in prepubertal IVEP programs may help improve oocyte competence.

## 5. Hormonal Stimulation

Due to the impaired HPO axis in prepubertal animals, an efficient hormonal stimulation protocol is critical to provide the COCs with a conducive intra-follicular milieu prior to LOPU. Previous work in our laboratory showed that FSH stimulation in prepubertal calves was able to mimic a functional HPO axis by increasing mRNA expression of FSH receptor (FSHR) and cytochrome P450 family 19 subfamily A member 1 (CYP19A1), while decreasing levels of steroidogenic acute regulatory protein (StAR) and hydroxy-δ-5-steroid dehydrogenase, 3β-and steroid δ-isomerase 1 (HSD3B1) in calf granulosa cells [62]. The molecular changes that occur during follicular and oocyte growth involving molecules synthesized within the oocyte or imported from granulosa cells are critical for the acquisition of an oocyte’s developmental competence and support the theory that “the history of the follicle determines the future of its oocyte” [92]. In support of this, several studies have shown a positive correlation between the follicular diameter and developmental competence of the oocyte in many species, including sheep [93], goat [94,95], cattle [35,72,92], buffalo [96], and pig [97,98]. For example, in adult cattle, oocytes from follicles 2–6 mm in diameter produced an average blastocyst rate of 34.3%, while oocytes from follicles > 6 mm in diameter produced an average blastocyst rate of 65.9% [72]. A similar pattern was observed in adult buffalo, with oocytes originating from follicles < 3 mm in size resulting in a blastocyst rate of 2.4 ± 1.5% while oocytes originating from follicles > 8 mm in diameter resulted in a blastocyst rate of 16.9 ± 1.7% [99]. This same trend was observed in prepubertal animals, with blastocyst rate per oocyte increasing from 6.8% to 13.8%, comparing oocytes from small (<5 mm) and large (≥5 mm) follicles in Holstein calves [100].

In prepubertal animals, LOPU-IVEP has been performed following hormonal stimulation protocols that were adapted from those used for adult animals. The goals of gonadotropin stimulation are not only to increase the size of follicles, and consequently oocyte competence, but also to increase the number of follicles suitable for aspiration [21]. Follicle stimulating protocols have consisted of multiple injections of FSH, single injections of compounds with a longer half-life such as equine chorionic gonadotropin (eCG), or a combination of both FSH and eCG [101,102,103]. Due to its short metabolic half-life, FSH is typically re-administered every 12 h for 3–4 days. Studies in the 1990s found that calves had a significantly better follicular response when subjected to multiple injections rather than a single injection of a large dose of FSH [102,103]. However, combining a single FSH injection with one of eCG resulted in a similar ovarian response to multiple FSH injections, suggesting a single dose of FSH is able to recruit but not sustain development of a follicle cohort [102,103]. These data seem to be supported by the fact that combining a single injection of FSH with a low dose of eCG can result in a similar ovarian response to multiple-injection regimes, with the FSH bolus able to recruit a follicle cohort, and the eCG able to sustain continued development [45]. It could be possible that eCG aids in follicle development from its inherent LH activity, which could act synergistically with FSH [103,104]. When comparing the interval between FSH, with and without eCG, we found that FSH injections every 8 h starting 72 h before LOPU, until a single dose of 400 IU of eCG 36 h prior to LOPU, yielded better blastocyst rates compared to FSH injections every 12 h without eCG (17.5 ± 8% vs. 8.9 ± 5%) [100].

## 6. LOPU and COC Quality

As the LOPU procedure is essentially the same for all ruminants and has been described in detail in other manuscripts [7,10,53], this review will not focus on the technical aspects of the procedure itself. However, it is worth highlighting that LOPU has been shown to be extremely safe and can be repeated on a regular basis. For example, LOPU has been repeated ~10 times in goats [20], and we repeated the procedure every two weeks in prepubertal Holsteins and buffalo between 6 and 9 times over a 3–4 month period [105]. Following this, none of the animals had reproductive problems later in life, as they were used to produce more embryos by trans-vaginal OPU and had normal fertility following artificial insemination. In our experience with prepubertal calves and buffalo, oocyte recovery rate (the proportion of follicles from which COCs were recovered) following LOPU is usually very good. Indeed, the average recovery rate was 77.1 ± 27% in Holstein calves (n = 109 LOPUs) [53], and 84.3 ± 29.3% in buffalo calves (n = 56 LOPUs, unpublished). Concerning COC quality, 87.4 ± 19% were deemed usable including 67% grade 1 and 20.4% grade 2 [53]. In addition, we observed that the gonadotropin stimulation regime used affected COC quality, with a longer stimulation protocol (≥72 h) resulting in a viability rate of 95.3% ± 18%, compared to 85.4% ± 22% for a shorter protocol (36–42 h) [53].

## 7. Individual Variation

In adult cows, the ovarian response upon gonadotropin stimulation is widely variable among animals [106]. The same variation was observed in calves [21,106], with research in our laboratory revealing similar results in both Holsteins [100] and buffalo [105] calves as shown in Table 1. The large individual variation is problematic in selecting the best calves to be used in a prepubertal LOPU-IVEP scheme, which may be mitigated by determining the serum concentrations of anti-Müllerian hormone (AMH) given its correlation with an individual animal’s response following gonadotropin stimulation observed in adult cattle and buffalo [107,108,109]. Although more work needs to be done to confirm this remains true in prepubertal buffalo calves, data suggest that AMH concentration remains a credible marker for LOPU-IVEP performance in prepubertal *Bos taurus* and *indicus* calves [110]. This is particularly useful since the follicular population is difficult to assess using ultrasound at such a young age.

### Seasonality

In the specific case of buffalo, another factor potentially contributing to variation in results is season. Buffalo are sensitive to long photoperiods, with reproductive efficiency improving in the autumn and winter as daylight decreases, similar to sheep & goat [111,112,113,114]. Season has been reported to influence the age at puberty [115]. Moreover, in adult Mediterranean buffalo undergoing repeated OPU, embryo yield improved significantly in the autumn [116], but there are yet no studies on the impact of season on prepubertal oocyte quality. Additionally, heat stress is well-researched and known to impact the estrous cycle, follicular development, oocyte quality and embryonic development rates in ruminants [117,118,119].

## 8. In Vitro Embryo Production

Following LOPU, oocytes undergo in vitro maturation, fertilization, and culture. Although variations exist in cattle and buffalo, these usually last for 22 h, 18 h, and 7 days, respectively. Most protocols have followed media compositions and procedures consistent with those used for adult animals with minimal derivations [23,101,120]. As such, commercially available media can be used. However, prepubertal oocytes may benefit from specially tailored IVEP protocols supplemented with various factors, which is discussed below.

### 8.1. Oocyte In Vitro Maturation (IVM)

Although in vivo maturation was the norm for many years, and the first Holstein calf born in the world from IVF was a product of in vivo maturation [121], in vitro maturation has yielded more reliable and consistent results in recent years. The objectives of IVM are both nuclear and cytoplasmic maturation. Nuclear maturation is the transition from germinal vesicle (prophase I) to metaphase II, while cytoplasmic maturation allows morphological, functional and biochemical changes to take place in the cytoplasm.

Multiple studies have shown that, although prepubertal oocytes are able to complete nuclear maturation, their ability to manage cytoplasmic maturation is more ambiguous. For nuclear maturation, it has been shown that oocytes can undergo germinal vesicle breakdown and successfully arrest at metaphase II [43,46,47,103]. It has been suggested that this process may be delayed in lamb oocytes compared to ewes [79]. However, our findings with oocytes collected from Holstein [53,100] and buffalo (unpublished) calves revealed that ~80% were able to mature to the metaphase II stage and successfully extrude the first polar body after 24 h of IVM. In terms of cytoplasmic maturation, electron microscopy studies have shown that organization of the oocyte organelles, such as the number and distribution of cortical granules as well as the population of mitochondria, are different in prepubertal compared to adult oocytes [47,79,122]. Damiani and colleagues compared cortical granule migration in calf and cow oocytes and found that cortical granules did not migrate as efficiently in calf oocytes as only 19% (17/90) of calf oocytes exhibited migration compared to 71% (83/117) in cow oocytes. This may impact normal fertilization and the initiation of the block to polyspermy, since 81% (73/90) of calf oocytes still possessed clusters of cortical granules following IVM [47]. Furthermore, cortical granule migration was delayed in 70% (19/27) of calf oocytes compared to 28% (7/25) in cow oocytes [47]. In addition to cortical granule migration, other cytoplasmic differences have been noted, including the distribution of mitochondria and lipid droplets [47]. These cytoplasmic deficiencies may be associated with the impaired competence of prepubertal oocytes. In support of this, it has been shown that transferring the nuclei of adult oocytes into enucleated calf oocytes resulted in similarly low development rates to those observed in control calf oocytes [123].

### 8.2. In Vitro Fertilization (IVF)

The ability of calf oocytes to properly manage fertilization, oocyte activation and the block to polyspermy appears to be impaired. Research in the 1990s showed that, although fertilization rates (as measured by sperm penetration) were the same between prepubertal and adult donors, there was a significantly higher rate of abnormal fertilization in prepubertal (16%) than adult (7%) oocytes [24,47]. Work in our laboratory provided additional evidence that polyspermy is a significant problem for IVF in calf oocytes. Working with Holstein calf oocytes and using the industry standard concentration of 1 million motile sperm/mL, polyspermy rates were over 40% [53]. However, when the sperm concentration was reduced to 500,000 motile sperm/mL, the incidence of polyspermy decreased to 19.7% [53]. In addition, the normal fertilization rate, as evidenced by the presence of two polar bodies and two pronuclei, increased from 59.4% to 69.7% [53]. Interestingly, we observed a steady decrease in polyspermy rates with age, declining from 45.5% in animals < 100 days old, to 12.8% in animals >130 days old [53]. We also observed similar results working with buffalo calves, with age and semen dose affecting polyspermy rates [105].

### 8.3. Embryo In Vitro Culture (IVC) and Transfer

Following fertilization, cell division appears to be delayed, with a low proportion of calf-derived embryos reaching the 4 and 8-cell stages of development at standardized time points [35,120]. In addition, embryo development to the blastocyst stage is significantly lower than what is achieved with adult Holsteins and buffalo oocytes [22,23,27,35,120]. In our experience with Holstein calves, cleavage rates varied between 60–70% and blastocyst rates were around 20%. However, both embryo yield and quality were significantly affected by the gonadotropin stimulation protocol and age of the calves [53]. In buffalo, this is potentially compounded by the fact that both the oocyte donor and sire used during IVF have a large influence on IVEP outcome, with only around 10% of males suitable for IVF [27,124]. Despite limited information in the published literature on the timing and causes of embryonic development arrest, it has been shown in 6–8-month-old heifers that 67% (40/60) of cleaved embryos that failed to reach the blastocyst stage arrested between the 2 and 8-cell stage, which was significantly higher than the 18% (5/28) observed in embryos from adult animals [120]. This suggests that prepubertal oocytes are unable to transition from oocyte to embryo and properly regulate embryonic genome activation, as the stage of developmental arrest coincides at around this time [125].

Recently, the possible impact of ARTs on the embryonic epigenome has garnered attention, with studies suggesting offspring produced by IVEP may be at higher risk of various disease [126,127]. For example, large offspring syndrome has been associated with epigenomic differences in imprinted genes [128,129]. Furthermore, the extent of cellular reprogramming and epigenetic inheritance of both parental methylomes on the embryo is currently being investigated [130]. Whether prepubertal LOPU-IVEP programs may affect epigenetic inheritance is unclear, however. Evidence in bulls suggest that the age of the sire influences the transcriptome and epigenome of blastocysts produced by IVF [131]. In females, transcriptomic comparison of blastocysts produced from the same heifers between 8–14 months old revealed that genes related to mitochondrial function were impacted in younger heifers [132]. How these differences may affect future embryo development of offspring is unknown.

Despite the lower development to the blastocyst stage, prepubertal embryos can reach this stage in a similar timeframe and have normal characteristics including a visible inner cell mass [120]. In terms of cell numbers, as an indicator of embryo quality, there were no differences in the trophectoderm-inner cell mass ratio between hatched and unhatched blastocysts from cows and 6–8-month-old heifers [120]. However, the total cell count in day 8 blastocysts was slightly lower but not statistically different between embryos of heifers (89 ± 20) and adult cows (100 ± 30) [120]. Additionally, heifer-derived and cow-derived blastocysts seem to have similar lipid metabolism, with day 8 blastocysts containing comparable triglyceride concentrations [120].

The ultimate and essential test for blastocyst quality is the ability to establish pregnancy and result in healthy offspring following embryo transfer. Pregnancies and live births with full-term offspring following LOPU-IVEP and embryo transfer have been reported by multiple authors using calf-derived oocytes in both Holsteins [23,43,44,53,101] and buffalo [7,27]. Although earlier studies have suggested lower rates of establishing pregnancy with prepubertal-sourced embryos, our findings revealed more encouraging results. Indeed, we obtained a 62% (13/21) pregnancy rate after transferring LOPU-IVEP blastocysts from Holstein calf oocytes. Of the 13 confirmed pregnancies, 4 were interrupted for experimental reasons and 100% of the 9 that were allowed to continue carried their pregnancy to term [53]. In buffalo, of 10 embryo transfers, 3 became pregnant, all of which delivered healthy calves [7,105]. Other authors reported similar results in prepubertal buffalo by confirming 3 pregnancies and delivery of healthy calves after the transfer of 8 IVEP embryos [133,134]. With the knowledge that these prepubertal LOPU-IVEP-ET schemes do work, animal breeding companies are now starting to offer these programs on a commercial basis. However, further research is needed to improve and ensure the long-term financial viability of these programs going forward.

### 8.4. Embryo Cryopreservation

In addition to yielding similar rates of embryos and pregnancies following transfer, another goal is for prepubertal-derived embryos to have cryotolerance similar to that of adult-derived embryos. It is well documented that in vivo produced embryos are more cryotolerant than their in vitro produced counterparts [135,136,137]. As such, embryo quality plays a major role in post-thaw survivability, with the cytoplasmic lipid content, i.e., the number and size of lipid droplets, shown to affect cryotolerance significantly, with more lipids being detrimental [135]. This presents a unique challenge for buffalo embryos, as they have high levels of lipids [3]. To address this problem, L-carnitine supplementation in vitro has been shown to aid in the lipid metabolism, as well as providing antioxidant protection, which improved post-thaw survivability in both Holsteins [138,139] and Buffalo [140,141]. However, this strategy remains to be tested in prepubertal-derived embryos.

## 9. Future Perspectives: What Can We Do Better?

With the knowledge that prepubertal LOPU-IVEP technologies do work, as evidenced by healthy calves born following embryo transfer, the current challenge is improving efficiency. As such, attention should focus on conditions both in vivo, before LOPU, and in vitro, following LOPU. In vivo approaches should include innovative gonadotropin stimulation protocols for young donor animals in order to enhance the intra-follicular environment and maximize oocyte development inside the follicle. In vitro approaches should focus on amending IVEP procedures to better accommodate the requirements of prepubertal oocytes to maximize meiotic maturation, normal fertilization and embryo development to the blastocyst stage.

### 9.1. Optimized Gonadotropin Stimulation

Efficient gonadotropin stimulation regimes should increase the size of follicles available for aspiration, as embryo development rates are directly associated with follicular size [35,72,92,96]. As such, gonadotropin stimulation over a longer period of time has been shown to be beneficial in calves. Work in our laboratory compared short (3 FSH injections, 12 h apart, starting 36 h prior to LOPU, total FSH 100 mg) vs. long gonadotropin treatments (6 FSH injections, 12 h apart starting 72 h prior to LOPU, total FSH 96–140 mg) and revealed that not only did the proportion of large follicles aspirated increase (11.2% vs. 34.0%), but cleavage rate (59.0 ± 23% vs. 72.7 ± 21%) and blastocyst rate (18.3 ± 15% vs. 36.7 ± 26%) were also significantly increased in the longer treatment [53]. Other studies have shown that an even longer stimulation duration of 7 days, compared to 4 days, resulted in a larger proportion (56.4 ± 8.3% vs. 27.8 ± 7.5%) and number (13.3 ± 1.8 vs. 9.0 ± 1.3) of large follicles (≥9 mm) [142,143]. However, the study focused only on the dynamics of follicular populations by serial ultrasound scanning, and the effects of such a prolonged protocol on oocyte competence and embryo development rates remains to be tested. Similarly, gonadotropin stimulation significantly increased the proportion of medium (4–8 mm) and large follicles (≥9 mm) in buffalo aged between 5 and 9 months [144].

### 9.2. Oxidative Stress and the Importance of Antioxidants

Oxidative stress caused by reactive oxygen species (ROS) can damage cells by disrupting homeostasis and leading to apoptosis. Glutathione (GSH) is considered the major line of defence against oxidative injury by helping to maintain the redox state within the cell. In addition to its role in preventing oxidative stress, GSH has been shown to play an important role in the transport of amino acids, as well as in DNA and protein synthesis [145]. The tripeptide thiol compound has been shown to be synthesised during oocyte maturation in bovine [146], bubaline [147], caprine [148], and porcine [149] oocytes. GSH is also known to play important roles in the formation of the male pronucleus and early embryonic development [150]. As oxidative stress is known to be pervasive during in vitro manipulation, compared to conditions in vivo, most IVEP protocols use antioxidants aimed at either promoting GSH synthesis (e.g., cysteine), or scavenging ROS (e.g., melatonin) [151]. Since oxidative stress is known to play a significant role in vitro and prepubertal oocytes may be deficient in their ability to combat ROS, it is plausible that they are more susceptible to oxidative stress [151,152]. As such, prepubertal IVEP may require specialized antioxidant treatments tailored to their needs. This may be especially important in buffalo because of the high concentration of lipids within the oocyte and therefore the increased risk of lipid peroxidation.

Although many different antioxidants have been tested and used over the years in adult IVEP schemes, there are fewer studies assessing the efficacy in prepubertal animals, especially in cattle and buffalo. Working with 1–2-month-old goats, Rodriguez-Gonzalez and colleagues found that IVM supplemented with cysteamine increased the GSH concentration, and improved blastocyst yield and total cell number per blastocyst [148]. Similar results were found in adult buffalo by Gasparrini and colleagues [153]. In a subsequent paper by the same group, they showed that supplementation with cysteamine combined with cystine, was even more advantageous than cysteamine alone, increasing the transferrable embryo rate from 23.8 ± 3.9% to 30.9 ± 5.8% [154]. Whether these findings can be applied to prepubertal animals remains to be determined.

Another antioxidant used in many IVEP schemes across multiple species is melatonin, which has been shown to reduce oxidative damage in the oocyte [155,156]. Melatonin is produced throughout the body, including the ovary, and has been detected in follicular fluid of bovine [157], porcine [158], bubaline [159], and caprine [155] follicles. In prepubertal goats, higher concentrations of melatonin were detected in large follicles (> 5 mm) compared to small follicles (<3 mm) [155]. The same trend was found in adult Murrah buffalo [159]. In prepubertal goats, melatonin supplementation during IVM increased the blastocyst rate [155], decreased intracytoplasmic ROS, improved ATP content, and enhanced mitochondrial activity [156]. Similar results were found in adult Holstein cows [160] and water buffalo [161]. While melatonin supplementation during IVM of COCs from 4–5-week-old lambs was found to have no effect on development rates [162], in 6–10-month-old Holsteins, it was shown to increase blastocyst rates from 11.1 ± 3.5% to 23.1 ± 5.1% [163].

### 9.3. Endoplasmic Reticulum Stress

Endoplasmic reticulum (ER) stress is a major contributor to embryonic death because physiological and exogenous stressors typically lead to disruptions in protein folding and ROS production in the ER [164]. Induction of ER stress has been shown to impair embryo development rates in multiple species [165,166], while ER stress inhibitors have been shown to improve IVEP development rates [165,167,168]. Tauroursodeoxycholic acid (TUDCA), a bile acid, was shown to inhibit ER stress and improve in vitro embryo development and blastocyst quality in different species [168,169,170,171,172]. TUDCA supplementation was shown to decrease the incidence of DNA double strand breaks in porcine blastocysts [168] and decrease intracellular ROS concentrations in oocytes from adult cattle [173]. In buffalo, treatment with TUDCA decreased cell apoptosis in embryos under ER stress induced by tunicamycin [166]. In prepubertal Holsteins, IVC supplementation with 50 μM TUDCA tended to increase blastocyst rates (30.9 ± 12% vs. 25.7 ± 2%) compared to the control [100]. More studies are needed to better evaluate the impact of TUDCA in prepubertal oocytes, such as testing higher concentrations during IVC. It is also possible that supplementing both IVM and IVC with TUDCA could further impact prepubertal IVEP because of its role in the regulation of calcium metabolism [174,175], which could also favor normal fertilization and embryo cleavage.

### 9.4. Cytokines and Growth Factors

Cytokines and growth factors are small peptide proteins involved in cellular signalling and communication. Fibroblast growth factor 2 (FGF2), leukaemia inhibitory factor (LIF), and insulin-like growth factor (IGF1) are among the growth factors found in follicular fluid that have regulatory effects on COCs. Working with porcine oocytes, Yuan and colleagues (2017) assessed the impact of adding these growth factors (in a cocktail coined ‘FLI’) to IVM media and observed a significant increase in oocyte maturation, embryo development and quality, and litter size following embryo transfer [176]. Working with lambs, Tian and colleagues found that combining FLI with insulin-transferrin-selenium (ITS) during IVM increased the blastocyst rate more than two-fold (44.2 ± 5.7% vs. 21.6 ± 4.6%) compared to the control group [162]. How these findings may benefit IVEP systems for prepubertal cattle and buffalo remains unknown.

### 9.5. Oocyte Pre-Maturation In Vitro

There is evidence that a short ‘pre-maturation’ period in presence of meiotic inhibitors such as c-type natriuretic peptide (CNP), epidermal growth factor receptor (EGFR) inhibitor, and cAMP prior to IVM may improve oocyte competence. During LOPU, separation of COCs from their follicles causes cAMP concentrations to decrease, resulting in spontaneous resumption of meiosis [177]. During pre-maturation, oocytes are temporarily arrested at the GV stage, to allow more time for cytoplasmic maturation to occur and promote synchrony among aberrant nuclear and cytoplasmic maturation [177,178]. Several studies have shown pre-maturation protocols able to increase blastocyst rate and quality [179,180]. CNP increases cGMP concentrations in COCs, which inhibits the cAMP hydrolyzing enzyme phosphodiesterase 3A, maintaining meiotic arrest [181]. Pre-maturation of prepubertal goats COCs for 6 h with CNP maintained TZP density, which is essential for cGMP transport into the oocyte and, consequently, meiotic arrest [182]. This treatment significantly increased blastocyst development rates compared to controls (30.2% vs. 17.2%), possibly due to an improved ability of the oocyte to manage oxidative stress, as CNP pre-maturation resulted in increased intra-oocyte glutathione concentrations and decreased ROS [182]. EGFR inhibition can also be used to reversibly arrest bovine COCs at the GV stage [183]. These pre-maturation protocols may represent a new alternative for use in combination with growth factors, antioxidants and inhibitors of ER stress to further improve prepubertal IVEP efficiency. However, these approaches require further validation.

## 10. Conclusions

Although several obstacles remain to be overcome, the use of prepubertal breeding schemes based on LOPU-IVEP is a powerful method for accelerating genetic gain. In Holsteins, the technology has reached a level of commercial viability, with several large biotechnology companies currently using this technology. Although the potential reward in applying this technology in buffalo is larger due to their prolonged sexual maturity, more work needs to be done for further efficiency optimization. Enhanced stimulation protocols yielding more competent oocytes at collection, coupled with in vitro procedures that will improve cytoplasmic maturation and the oocyte’s machinery to fight oxidative and ER stress, are among the improvements that will likely increase the proportion of competent oocytes recovered from prepubertal compared with post-pubertal animals.

## Figures and Tables

**Figure 1 animals-11-02275-f001:**
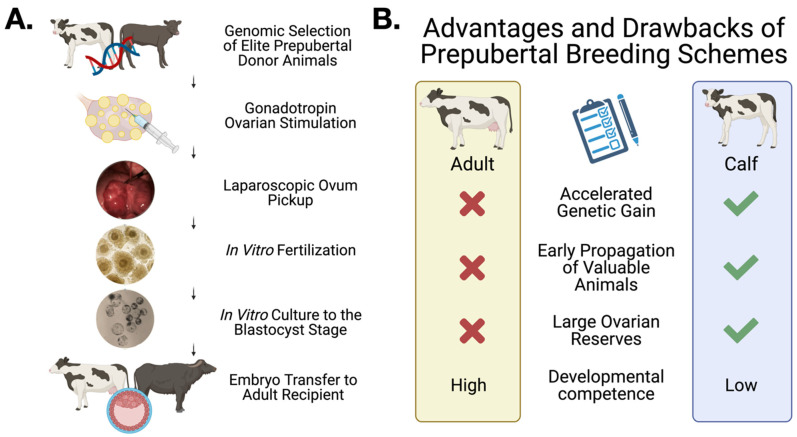
Overview of LOPU-IVEP in Prepubertal Buffalo and Cattle. (**A**): Flow chart showing the typical steps involved in prepubertal LOPU-IVEP programs. (**B**): Comparison between adult and prepubertal breeding schemes, showing the advantages and drawbacks of each. Figure created with BioRender.com, accessed on 30 July 2021.

**Figure 2 animals-11-02275-f002:**
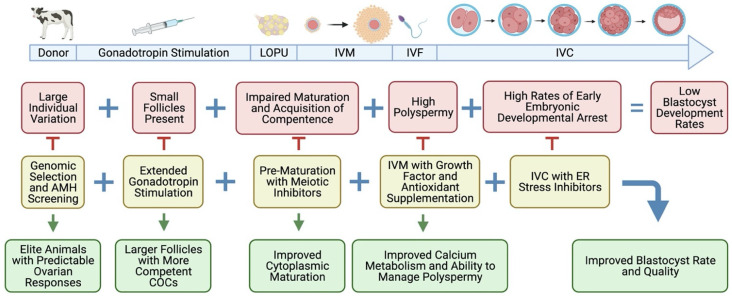
Overview of LOPU-IVEP in prepubertal cattle and buffalo, showing potential targeted approaches to address the main problems holding the technology back. Figure created with BioRender.com, accessed on 30 July 2021.

**Table 1 animals-11-02275-t001:** Individual variation of usable COCs recovered from calves over six LOPU sessions.

Species	Number of Animals	Number of COCs Recovered			
		Total	Mean ± SD All Calves (Total Per Calf)	Mean ± SD Bottom Calf (Total)	Mean ± SD Top Calf (Total)
Holstein	11	1393	22.2 ± 14 (126.6)	12.7 ± 4 (72)	38.2 ± 11 (229)
Buffalo	8	774	16.2 ± 9 (81)	10.1 ± 3 (50)	26.6 ± 6 (130)

SD = standard deviation. Data adapted from [100,105].

## Data Availability

No new data were created or analyzed in this study. Data sharing is not applicable.

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
