# Peer review of "In Vitro Production of Embryos from Prepubertal Holstein Cattle and Mediterranean Water Buffalo: Problems, Progress and Potential"

_animals, 2021, doi:10.3390/ani11082275_

Round 1

Reviewer 1 Report

The authors prepared a very nice review about IVF  using oocytes obtained from prepubertal heifers, considering dairy cattle and buffalo. The language is concise and clear. Over 160 important references were included. The text covers very important topics, like i) history; ii) oocyte competence; iii) hormones; iv) individual variation; v) the IVF technique; vi) embryo transfer, and others. 
I have two minor points:
Authors prepared the title 'dairy cattle'. However, the main focus seems to be only Holstein. In my experience, there are some important differences involving Jersey and Holstein breeds, considering oocyte collection, IVF production, etc. I kindly invite the authors to include more information about Jersey and/or making clear the title. 
The second point is about the reproductive seasonality of the buffalos. In this species, the mature females may present a well-established period for reproductive activity, according to the latitude.  Despite the review is focused on prepubertal heifers, I understand the topic of seasonality should be commented on/improved. 

Author Response

Thank you for taking the time to review our manuscript and providing your valuable insight. Regarding your comment about the differences between Holsteins and other breeds of dairy cattle, we have adjusted the title of the manuscript to clarify that the focus of the manuscript is specific to Holsteins. Your comments about reproductive seasonality in buffalo were excellent and a very important topic we should have mentioned, we have therefore added a paragraph discussing this.

Reviewer 2 Report

The article of Currin et al. Corresponds to a bibliographic review focused on In Vitro Production of Embryos from Prepubertal Dairy Cattle and Buffalo: Problems, Progress and Potential. The topic is interesting and novel, providing recent information regarding new strategies to improve the in vitro embryo production from prepubertal dairy cattle and buffalo.
To complete the interesting review I suggest adding another two topics to this review if possible:
First topic: about the practical application of embryo transfer produced from LOPU and IVEP  in commercial farms and success rates.
Second topic: cryopreservation of embryos produced from LOPU and IVEP, I think that this is also a big challenge to success the embryo transfer after in vitro production that needs to be mentioned in this review.

Author Response

Thank you for your important comments and taking the time to review our manuscript. Concerning the practical applications of prepubertal LOPU-IVEP-ET on commercial farms, we have added some information to better discuss this topic in the manuscript, as big semen/animal breeding companies now have their own JIVET programs and are starting to offer their services on a commercial basis. Your comments about cryopreservation and tolerance were very beneficial, so we have added a paragraph to the manuscript discussing this.

Reviewer 3 Report

Dear Editors,

The manuscript “In Vitro Production of Embryos from Prepubertal Dairy Cattle and Buffalo: Problems, Progress and Potential”, as the title indicated, describes the main advances on ARTs using prepubertal animals. The manuscript is really well written and is gathering the most valuable and updated information regarding this subject. This work will be a great reference for anyone seeking knowledge about the prepubertal reproductive approach. A minor question and one suggestion are added in order to conclude this review.

-Figure 1. I would suggest displaying letters (A and B) to better explain the two topics.

-The authors described greatly the genetic gain by shortening the reproductive time. Nonetheless, do the authors know if using prepubertal animals could bring any impact on the epigenome? It is well known that artificial reproductive technologies are prone to display epigenetic disorders but using this approach could also be an important tool to study epigenetic inheritance. If available, please add some references about it in the manuscript.

Author Response

Thank you for your feedback and reviewing our manuscript. Your comments were valuable and have made edits to address your observations. We have added ‘A’ and ‘B’ panels to Figure 1 to clarify the two elements. Thank you for your comments about the possible impacts that prepubertal LOPU-IVEP-ET schemes may have on epigenetics. We have added a short paragraph and some recent references to address and discuss this important emerging topic.